# AI University: An LLM-Powered Learning Assistant for Engineering—A Finite Element Method Case Study

## Abstract

We introduce AI University (AI-U), a flexible framework for AI-driven course content delivery that adapts to the classes' instructional styles. At its core, AI-U combines a fine-tuned large language model (LLM) with retrieval-augmented generation (RAG) and a reasoning synthesis model to generate instructional style-aligned responses from lecture videos, notes, and textbooks. Using a graduate-level finite-element-method (FEM) course as a case study, we present a scalable pipeline to systematically construct training data, fine-tune an open-source LLM with Low-Rank Adaptation (LoRA), and optimize its responses through RAG-based synthesis. Our evaluation—combining cosine similarity, LLM-based assessment, expert review, and user studies—demonstrates strong alignment with course materials. We have also developed a prototype web application, available at (link removed for anonymous submission), that enhances the instructional content of the AI-generated responses with references to relevant sections of the course material and clickable links to time-stamped instances of the open-access video lectures. Our expert model is found to be higher scoring by a quantitative measure on 86% of test cases. An LLM judge also found our expert model to outperform state-of-the-art open source models approximately four times out of five. Human evaluation by advanced users showed a preference for our expert model approximately twice as often as for the competing open-source model. The FEM course instructor found our expert model to achieve better alignment with class-specific content than a recent closed-weight model when each was combined with the reasoning synthesis model. AI-U offers a scalable approach to AI-assisted education, paving the way for broader adoption in higher education. By presenting our framework in the setting of a class on FEM—a subject that is central to training PhD and Master students in engineering science—we offer a template with potential for extension across STEM fields.

## 1 Introduction

Large language models (LLMs) have risen to positions of dominance across a wide range of applications, shaping everyday interactions with artificial intelligence. Most widely available LLMs are trained on an extensive corpus of internet-sourced data, enabling them to achieve remarkable accuracy in general tasks. However, there is a growing need for domain-specific LLMs tailored to specialized knowledge areas. As LLMs continue to evolve, the next frontier lies in developing models optimized for task-specific roles. This shift is further reflected in the emergence of agentic workflows, where multiple specialized models collaborate to complete complex tasks. Thus, developing task-specific LLMs becomes essential. In this work, we contribute to this trend by tailoring a model to the unique instructional style and learning objectives of a specific university course in engineering science.

Fine-tuning LLMs for domain-specific applications offers several key advantages. Industries can train these models on proprietary data, ensuring they align with specialized knowledge and business needs while also adopting a desired style and behavior. Additionally, fine-tuning allows for the incorporation of new information beyond the original training cutoff, which is particularly crucial as LLMs have now reached a stage where they encompass vast internet-based knowledge (Ding et al., 2023). Rather than training models from scratch, fine-tuning provides a more efficient way to update and refine them with the latest data. Furthermore, this

approach enables the creation of highly personalized AI assistants tailored to the style of individual users, enhancing adaptability and user experience.

These benefits extend to the classroom, where there is a growing demand for scalable, accurate, and interactive teaching aids that support educators and enhance student learning while taking into account student privacy and equitable access. Here, we propose a structured framework, *AI University (AI-U)*, designed for university courses and adaptable to instructors' needs. We apply this framework to a graduate-level course on the Finite Element Method, a numerical technique for solving partial differential equations (PDEs) that is a widely used computational framework for engineering simulations (see Section "The Finite Element Method" for background). In particular, we aim to mirror the style of a course taught over multiple semesters, with recorded lectures available online. While existing LLMs have a broad understanding of the subject matter based on publicly available knowledge, they lack the distinct instructional style of this course. This includes the instructor's approach to introducing new concepts, the preferred use of terminology, symbols and mathematical techniques, and their unique conversational tone for instruction. To address this gap, we propose a platform that integrates LLMs with retrieval-augmented generation (RAG) to create a customized AI assistant tailored to the course. We use the evaluation metrics of cosine similarity and LLM-as-a-judge in conjunction with human evaluation by domain-specific experts to demonstrate the effectiveness of AI-U.

The current study follows a static approach, where all course materials are available before fine-tuning the LLM. However, the workflow is designed to be dynamic, allowing instructors to fine-tune the model with initial course materials at the start of the teaching term and then continuously update it with new lecture notes, or other content, through a RAG-based synthesis model with reasoning. The fine-tuning data is generated through a pipeline that takes course materials and produces question-answer pairs used to fine-tune a domain-expert LLM, which we call LLaMA-TOMMI-1.0 (Trained On Mechanics Materials Instructor). By combining responses from LLaMA-TOMMI-1.0 with real-time retrieval of, and reasoning on, course-specific information, our approach ensures that the assistant remains up-to-date and oriented with the evolving content of the course, ultimately creating a more adaptive and personalized learning experience. We note that, while proprietary models were explored for some portions of this work, the final workflow supports entirely open-source resources and models, supporting local hosting for data privacy (Dorca Josa & Bleda-Bejar, 2024) as well as equitable access to all learners.

Overall, AI-U represents an advance in integrating AI into education, enhancing both instructional efficiency and, potentially, student engagement. Its main contributions include:

- A scalable AI-driven question-answer data generation pipeline to produce a domain-specific fine-tuning dataset, with outputs verified by domain experts.

- A workflow in which a fine-tuned expert model, LLaMA-TOMMI-1.0, feeds into a RAG-based reasoning model for synthesis, enabling adaptable data updates and the generation of responses in the style of the course with course-specific references.

- A prototype web application centered on RAG and a reasoning synthesis model that integrates AI-generated responses with relevant course materials and open-access video lectures playable at the related timestamps.

- A pipeline demonstration using a fine-tuned open-source model; additionally, the entire system can easily be built with open-source tools, enabling local deployment and reducing privacy risks from external data sharing.

- Our dataset and code are available on Huggingface[1] and GitHub[2].

---

[1] Link removed for anonymous submission

[2] Link removed for anonymous submission

## 2    Related Work

### 2.1    Large Language Models

Modern LLMs can be traced to the seminal work by Vaswani et al. (2017), its focus on the attention mechanism and its ability to help scale training. This has led to the general trend of "bigger-is-better," with an emphasis on more training data and larger models. Although proprietary models, such as the GPT series, have until recently topped benchmark tests, the performance of open-source models such as BERT, LLaMA (Grattafiori et al., 2024), and DeepSeek (Bi et al., 2024; Liu et al., 2024) has steadily improved. It has become commonplace to fine-tune a pre-trained base LLM for applications requiring domain-specific information. The understanding that over-parameterized models reside on low intrinsic dimension (Aghajanyan et al., 2020; Li et al., 2018) led to Low-Rank Adaptation (LoRA) (Hu et al., 2021). LoRA is a parameter-efficient fine-tuning method that adds low-rank matrices into the frozen layers of a pre-trained model. Additional references or proprietary information can be provided to the base LLM through methods such as RAG (Lewis et al., 2021). At a basic level, a RAG pipeline will take an input sequence, embed it using an embedding model, and use a pre-trained *retriever* to find the top-k most relevant documents. These documents are typically embedded offline using the same embedding model and stored for later use. The original query is then *augmented* by the retrieved documents and used by a *generator*, typically an LLM, to produce the final response.

### 2.2    Applications in Education

As generative AI technology has evolved, so too have examples of their use in higher education settings (Chevalier et al., 2024; Wang et al., 2024; Watterson et al., 2025; Xu et al., 2024a;b; Zerkouk et al., 2025). These include their use as debugging tools for computer science students (Yang et al., 2024), simulating a classroom environment for users (Zhang et al., 2024), generating questions for students to test their knowledge Witsken et al. (2025) and serving as a teaching assistant (Hicke et al., 2023; Anishka et al., 2024). We especially highlight the work by Hicke et al. (2023), combining a LLaMA-2 base model with supervised fine-tuning, RAG and Direct Preference Optimization (DPO) to create an AI teaching assistant for an introductory computer science course. Notably, their training data source consists of available question-answer pairs from eight previous course semesters, which is not available for our course. Instead, our data workflow will enable instructors to fine-tune a course assistant when historical data is not available, while also delivering a platform whose functioning mirrors the course's instructional style.

### 2.3    Shortcomings and Concerns on LLM use in Education

The advances notwithstanding, there remain concerns preventing the rapid adoption of LLMs in the classroom. Commercial and third-party software bring concerns about data privacy and security, both for student data as well as proprietary teaching material (Chan, 2023). Equitable access is another concern, with students of higher socioeconomic status or AI literacy appearing to benefit the most (Yu et al., 2024). To encourage students to use them over commercially available options, course-provided assistants will need to be tailored for course-specific terminology, materials, and teaching styles. RAG-based approaches, while relatively simple to implement, are limited by their context windows. While this has improved greatly with recent LLMs, they still lack the ability to tailor a response based on a large corpus of reference data. Xing et al. (2024) showed that knowledge graphs are one effective approach to improving scalability and performance of RAG-based systems.

### 2.4    The Finite Element Method

The finite element method is a numerical technique to solve partial differential equations (PDEs) (Hughes, 1987; Zienkiewicz et al., 2013). It leverages variational methods to convert a PDE from the strong form to the weak form. The smoothness and differentiability requirements on the solution are relaxed by multiplying the PDE with test functions and transferring spatial derivatives to these test function. Infinite dimensional PDEs are furthermore reduced to finite-dimensional weak forms by introducing basis functions, which are

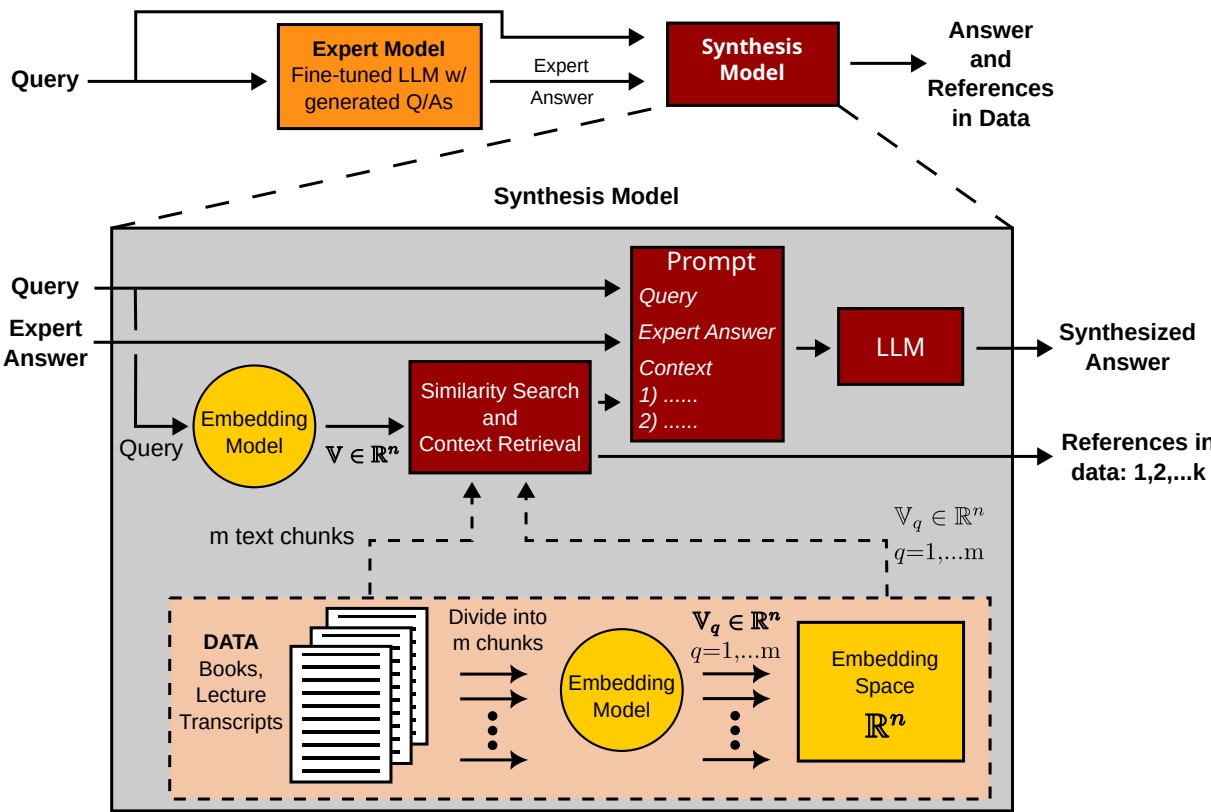

Figure 1: Overview of the AI University framework. Sections marked with a dashed line are performed once, at the beginning.

typically piecewise polynomials defined over finite subsets (elements) of the spatial domain. Time-dependence is usually treated by discretizing the time interval of interest into a finite number of sub-intervals and using integration schemes. These steps enable approximate, finite-dimensional solutions to PDEs with controllable, rigorously provable stability, accuracy and rates of convergence.

## 3 Methods

### 3.1 Inquiry Pipeline

The inquiry pipeline is shown in Figure 1. A high-level summary is provided here, with additional details in the sections below. The user's query is first answered by an expert model, a fine-tuned LLM trained to respond in the instructional style specific to the course. This expert-generated response, along with the original user query, is then passed to a synthesis model via a carefully constructed synthesis prompt. Within the synthesis pipeline, relevant context is first retrieved using the query through a RAG-based pipeline from a database of embedded course materials. Next, leveraging the synthesis prompt, an LLM integrates the expert model's response with the retrieved context to produce an enhanced, unified answer. This synthesized response includes relevant reference links pointing directly to specific sections of the course materials, including video lectures.

### 3.2 Training Data

The training data generation pipeline is shown in Figure 2. We approach the generation of training data in a manner similar to an instructor preparing course materials, by starting with the reference textbook for the course. In this case, we choose a canonical work in the field Hughes (1987). With the author's permission,

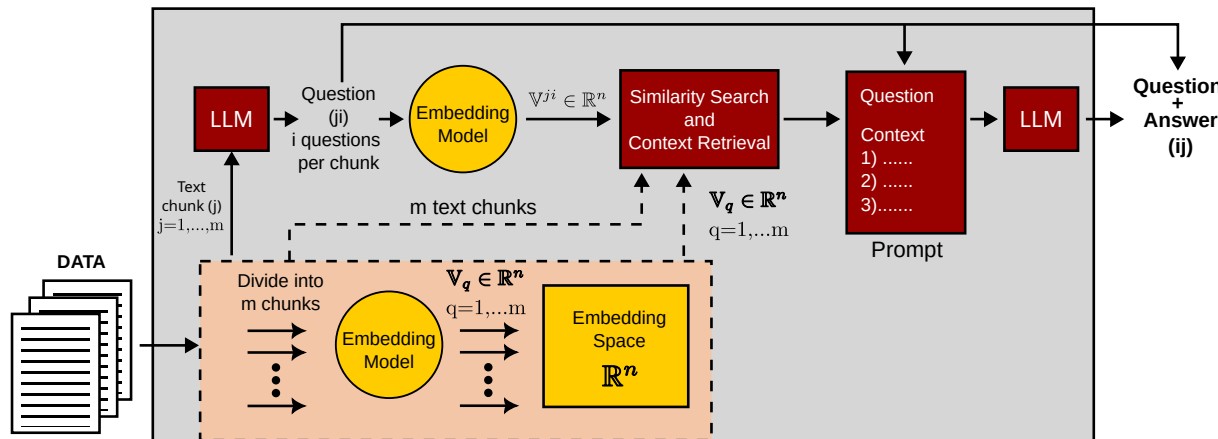

Figure 2: Overview of the data generation framework. Sections marked with a dashed line are performed once, at the beginning.

we convert textbook PDF files to LaTeX using commercially-available document conversion software[3] with minimal intervention (e.g., adding section headings and fixing formatting errors in equations). We estimate that the average time spent per chapter of the book was one human-hour. We leverage all material from the textbook, including exercises and examples, with the exception of the images. Subject-matter-expert audits of the output and restructuring of the LaTeX sections ensure data integrity.

In addition to the textbook, we leverage course content with permission of the content creators. We obtain transcripts (automatically generated) of the course lectures from the online playlist, from which we generate a series of question-and-answer pairs using ChatGPT 4o with the same workflow and prompts as the textbook material.

We meticulously generate question-answer pairs for each reference section using an LLM and the following carefully architected workflow (Figure 2): The reference material is divided by its natural boundaries—e.g., textbook sections and video segments. Using prompt engineering to achieve the desired format and focus, we generate as many questions as necessary to cover the material in each section ("chunk") of reference material (Technical Appendix, Section B.1 "Textbook question generation" for prompt). For each subsequent question, we retrieve reference material chunks from the available course material using cosine similarity between the embedded prompt and reference material. Through prompt engineering, we ensure that we only use the supplied reference material during answer generation and not the LLM's background knowledge of FEM (see Technical Appendix, Section B.1 "Textbook question generation" for prompt). We used ChatGPT 4o for this study, because of its well-documented performance at multimodal comprehension (OpenAI et al., 2024; Shahriar et al., 2024). Any other LLM could be used; however, the performance would differ and would require careful testing.

We incorporate additional course-related coding assignments as training data to enhance the dataset. We systematically generate question-answer pairs from existing course assignments using ChatGPT 4o. To ensure sufficient data coverage, three distinct prompting strategies are employed. The first prompt aims to generate a comprehensive set of question-answer pairs with detailed coverage of key concepts. These pairs are derived from coding assignments that provided a code template requiring modification (Technical Appendix, Section B.4 "Coding question-answer generation prompt 1"). The second prompt focuses on creating question-answer pairs that mirror the conceptual depth of the original coding assignments, ensuring that the questions accurately assess the same knowledge areas (Technical Appendix, Section B.5 "Coding question-answer generation prompt 2"). Since students have the flexibility to code in both C++ and Python, we introduce a third prompt to generate question-answer pairs specifically tailored for Python, aligning with the structure and intent of the C++ assignments (Technical Appendix, Section B.6 "Coding question-answer generation prompt 3").

---

[3]https://mathpix.com/

In total, 4648 question-answer pairs are generated from the data sources (textbook, 1053; coding examples, 286; online course transcripts, 3309). We conducted human expert reviews of 200 of these question-answer pairs, evaluating them against two criteria, as follows. Criterion 1: The technical correctness of the question and answer, their completeness, accuracy and scientific depth, followed by their relevance to this specific FEM course. Criterion 2: The grammatical and semantic quality of the question and answer. Both criteria were scored on scale of 0 - 3, where scores of 3 were perfect from all of the above perspectives. This resulted in a mean score of 2.64 on Criterion 1 with a std dev = 0.175, and mean of 2.74 on Criterion 2 with a std dev = 0.122. Of the 200 question-answer pairs, only 14 (7%) received a score of zero on one or both criteria. A more detailed analysis appears in Technical Appendix, Section C. On this basis and standard-establishing precedents in the literature (Northcutt et al., 2021) the question-answer pairs were found to be of sufficient quality for training. We reserve 10% of this data for testing, using the rest for training and hyperparameter optimization.

### 3.3 Fine-tuning Expert Model

To fine-tune an expert model, we select Llama-3.2-11B-Vision-Instruct (Grattafiori et al., 2024) as our base model due to its balance between performance and computational efficiency. While larger models may offer superior general knowledge, a medium-sized model like Llama-3.2-11B provides strong domain-specific capabilities with reduced resource overhead.

We perform fine-tuning within the Hugging Face ecosystem using the Transformers library (Wolf et al., 2020). This leverages PEFT (Parameter-Efficient Fine-Tuning) (Mangrulkar et al., 2022) to implement LoRA, enabling efficient adaptation of the model without modifying all parameters. Chat templating is handled using the `PreTrainedTokenizerFast.apply_chat_template` method. For domain adaptation, we employ the system prompt outlined in the Technical Appendix, Section B.3 "Fine-tuning system prompt". Model weights are loaded in bfloat16 for half-precision computation. We utilize the Accelerate library (Gugger et al., 2022) to distribute fine-tuning across two A40 GPUs, optimizing memory usage and training speed.

#### 3.3.1 Hyperparameter optimization

We employ Optuna for hyperparameter optimization, to search for optimal configurations to enhance model performance (Akiba et al., 2019). The following hyperparameter space is explored:

- **Learning rate:** varied in the range $[1e{-}5, 1e{-}3]$
- **Gradient accumulation steps:** 2
- **Epochs:** 5
- **LoRA parameters:**
  - **Rank:** varied in the range [8, 64]
  - **Alpha:** varied in the range [32, 128]
  - **Dropout:** chosen from [0.05, 0.1]
  - **Target modules:** {q, k, v, o, gate, up, down}
- **Warmup steps:** 100
- **Max token length:** 700

Optimization is conducted on two A40 GPUs, utilizing the full training dataset with cross-entropy loss. After hyperparameter tuning, the optimal hyperparameters obtained are: LoRA Rank = 45, LoRA Alpha = 65, LoRA Dropout = 0.05, and Learning rate = $5e{-}5$. Training logs and experiment tracking are managed via Weights and Biases (WANDB) to ensure reproducibility and analysis of model performance across trials[4]

---

[4]link removed for anonymous submission

### 3.4 RAG and Synthesis Model

We use a RAG-based pipeline to retrieve course-specific material. A user query is embedded into the same space as the course content, and relevant chunks are identified using cosine similarity. The top-k reference chunks are then returned to the synthesis LLM, along with the LLaMA-TOMMI-1.0 response (Figure 1). The synthesis model operates under a detailed system prompt (Technical Appendix, Section B.7 "Synthesis model"), which has been carefully refined to enforce a multi-step synthesis algorithm. Executing such a complex chain of instructions reliably is challenging for standard LLMs of small to medium parameter sizes, which often excel at fluency but struggle with strict logical constraints. To address this, we employ DeepSeek-R1-0528-Qwen3-8B as our default synthesis model. The advantage of using a dedicated reasoning model like DeepSeek-R1 over a general-purpose LLM of a similar size stems from its specialized training regimen. These models are typically fine-tuned not just on vast web data, but on curated datasets rich with structured reasoning paths, mathematical problem-solving, and code generation. This training encourages the model to internally decompose complex prompts into a sequence of executable steps, akin to a program, rather than treating them as stylistic suggestions. For our application, this is critical. For instance, our prompt first mandates a conditional check: the model must analyze the initial expert answer and retrieved context and, if the information is collectively insufficient, halt and output a specific instruction to the user. We qualitatively note that DeepSeek-R1 excels at this type of structured, procedural execution compared to Llama-3.2-11B-Vision-Instruct or GPT-4o-mini.

If the check passes, the model proceeds to a more nuanced reasoning task dictated by our guidelines: it must evaluate the quality of the expert's answer. If the answer is strong, the model must use it as a foundation and meticulously enrich it with details and citations from the retrieved context. Conversely, if it is weak or inaccurate, the model must pivot to rely more heavily on the retrieved material and construct an answer. Throughout this process, it must strictly ground its response in the provided course-specific sources and ignore its existing parametric knowledge. Additionally, the platform outputs the reasoning model's thinking process in which it analyzes the initial expert answer by comparison with the video lectures and textbook as sources. The visibility of the thinking mode and corrections to the initial expert answer can enhance the learner's understanding. To build further confidence in the final response, we prompt the synthesis model to label its response with one of three confidence levels: High, Medium, or Low. The model also generates text explaining the rationale for its chosen confidence level. Furthermore, it must meticulously apply formatting rules, such as generating precise Markdown, including appropriate citations to retrieved data, and correctly handling both inline and display LaTeX equations. This ability to faithfully execute a complex logical workflow makes the reasoning model significantly more reliable and capable for our high-fidelity synthesis task than models without reasoning. Demonstrations of the platform's output including the initial expert answer, the retrieved context from video lectures and the textbook, and the reasoning model's thinking process, are available as Supplementary Materials.

### 3.5 Evaluation of Fine-Tuning Effectiveness by Cosine Similarity & LLM-as-a-Judge

We hold out 10% of the training data for testing. The base model and LLaMA-TOMMI-1.0 are queried with the test questions, and their responses are recorded. Two approaches are presented here to evaluate the effectiveness of our fine-tuning.

First, we embed the model responses and use cosine similarity to evaluate how semantically similar they are against the embedded answers generated from the training data generation pipeline (defined as "ground truth"):

$$\text{cosine similarity} = \frac{r_{emb} \cdot \hat{r}_{emb}}{\|r_{emb}\|\|\hat{r}_{emb}\|} \tag{1}$$

where $r_{emb}$ is the embedded ground truth answer and $\hat{r}_{emb}$ is the embedded model response. We report both the average cosine similarity and win rate across all test data, where win rate is defined as the number of times a given model has the higher cosine similarity when answering a test question.

Next, we use an independent "LLM-as-a-judge" to evaluate the effectiveness of fine-tuning for adopting the course style. The ground truth reference embedded answers, base model response, and LLaMA-TOMMI-1.0 response are provided to the judge with one of two system prompts. The first prompt evaluates lexical

matching, structural similarity, and example consistency, returning the winning response based on these form-oriented criteria. The second prompt evaluates content accuracy, conceptual alignment, and completeness, returning the winning response based on these accuracy-focused criteria. The full prompts are provided in the Technical Appendix section "LLM-as-a-judge". In both cases, the judge is instructed to return either the base model as winner, LLaMA-TOMMI-1.0 as winner, both models if equally aligned, or neither model as being aligned with the ground truth response. The results of this evaluation, reflecting the performance of the fine-tuned model, are presented in Table 1. We acknowledge that some bias may be introduced by using ChatGPT 4o to both generate the question-answer training pairs and act as the judge. It is for this reason that we first provided the cosine similarity metric. Furthermore, we followed up with multiple human user evaluation studies described next, as further justification.

### 3.6 Human Evaluation of Framework Responses

Motivated by the intended classroom use of the AI-U framework, we also conducted human evaluations of its responses. Numerous model and/or platform combinations were evaluated by four users—all authors of this work—who are advanced ("advanced user") in the subject (e.g., postdoctoral researchers). In all cases, the output order was randomized and masked to preserve blinding. For head-to-head comparisons, four choices were allowed for each evaluation, with the response based on technical correctness and usefulness in gaining a deep understanding of the course material: (1) Model 1 (randomly labeled) is better; (2) Model 2 (randomly labeled) is better; (3) The models perform equally well; (4) The models are equally wrong.

In the first evaluation, the base and fine-tuned models were given the same set of 100 questions that students had asked the instructor in a previous semester of the course. Next, a set of 80 questions—also from students in a previous semester—were posed to the RAG and synthesis pipeline with DeepSeek-R1 as the synthesis model (herein referred to as the 'Platform'). The advanced users rated the responses of the two LLMs for technical correctness and usefulness in gaining a deep understanding of the course material both without and with the Platform. In a final evaluation of the base LLaMA-3.2-11B and LLaMA-TOMMI-1.0 models, the FEM course instructor also evaluated the two LLMs' performances on 60 questions on the Platform. This instructor evaluation was aimed at *discerning the alignment of the two LLMs with the instructional style of the course.* After completion of all human evaluations by both groups of reviewers, the model identities were decoded to tally the human evaluation results, which appear in Table 2.

To further assess the capabilities of LLaMA-TOMMI-1.0 as compared to other stand-alone models, an expanded analysis of three different expert models was performed The responses for each of three models—GPT-5, DeepSeek-R1-0528-Qwen3-8B, and LLaMA-TOMMI-1.0—were evaluated on a combination of technical accuracy and completeness. Each response to the same set of 80 questions previously used was given a point if the response was deemed "sufficient." The performance is reported in Table 3.

As a final evaluation, we performed two comparisons of the final platform, again using the same eighty questions as before. First, we compared (a) OpenAI's GPT-5 against (b) LLaMA-TOMMI-1.0 with the Platform (Table 4). In the second evaluation, we compared the Platform using two different expert models: (a) OpenAI GPT-5 against (b) LLaMA-TOMMI-1.0 (Table 5). The course instructor also evaluated these responses to discern alignment with the instructional style of the course.

### 3.7 Web Application

Figure 3 presents the user interface demonstration, available at (link removed for anonymous submission). The current implementation uses the Streamlit app[5] and is hosted on the HuggingFace spaces platform[6]. Users can select the number of relevant video lectures and textbook sections to retrieve, along with the maximum number of context tokens allowed per content. The platform's architecture is modular, allowing users to select different models for both the expert and synthesis roles. The default models represent our recommended configuration, fine-tuned and adjusted via our system prompts to ensure that the platform performs as designed. Alternate models are also available to accommodate user preferences and allow for

---

[5]https://streamlit.io/
[6]https://huggingface.co/spaces

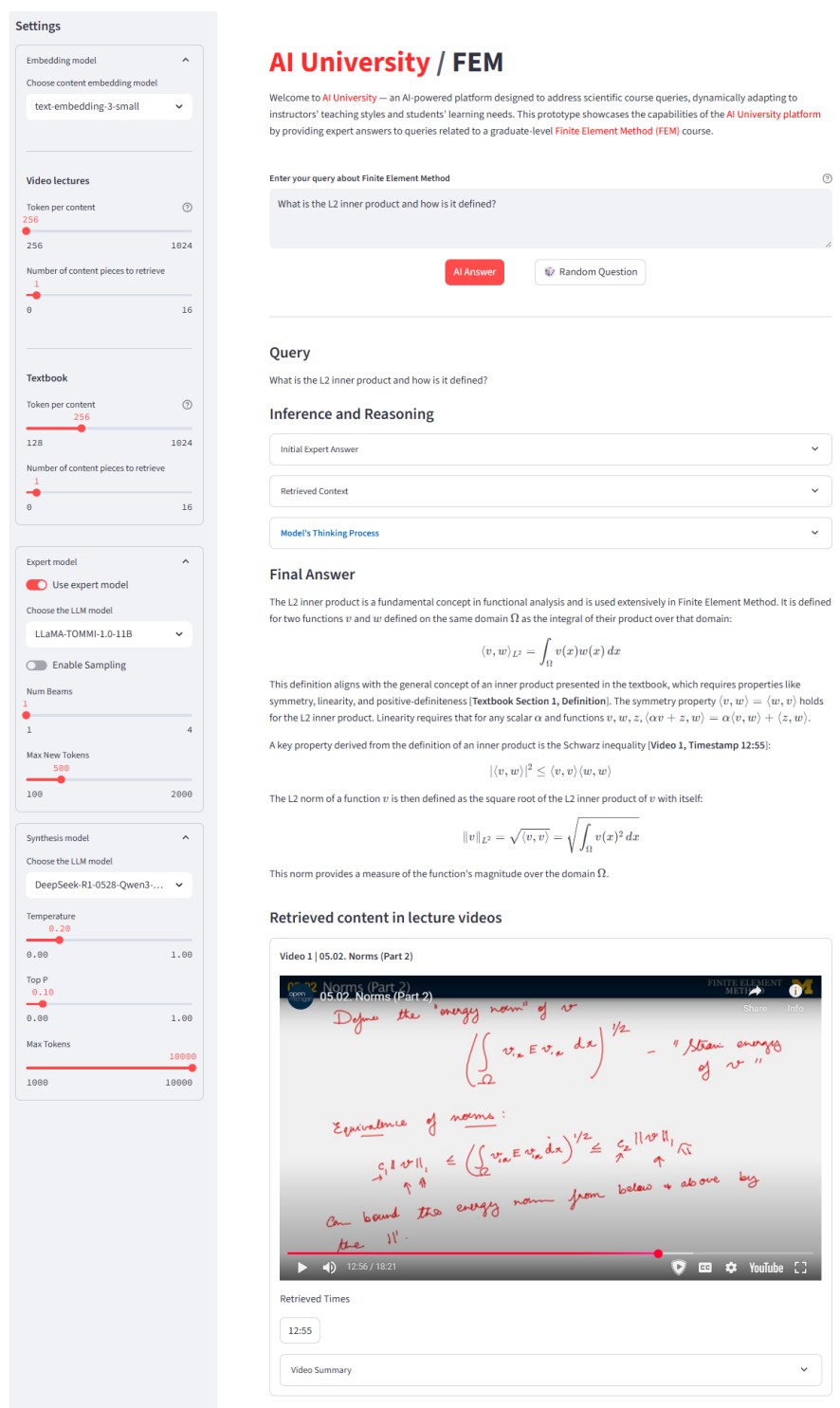

Figure 3: Demonstration of the web application, available at (link removed for anonymous submission).

experimentation. For the expert model, the default is our fine-tuned LLaMA-TOMMI-1.0. Users have the option to switch to the open-source LLaMA-3.2-11B or the commercial GPT-4o-mini; these alternatives are not fine-tuned and guided solely by prompt engineering. The expert model's generation can be configured by choosing a strategy (beam search or sampling) and setting parameters such as temperature, Top P, and

Table 1: Fine-tuning effectiveness, evaluated using cosine similarity and LLM-as-a-judge. For cosine similarity, both the ground truth label and the model response are embedded using OpenAI's latest vector embedding model (text-embedding-3-large) and used to calculate the average cosine similarity across all results ("Average Cos. Sim."). When choosing which answer is best aligned using cosine similarity ("Winner Cos. Sim."), the results show an overwhelming preference for the LLaMA-TOMMI-1.0 model. Two prompts were provided for the LLM-as-a-judge evaluations, (Technical Appendix, Section "LLM-as-a-judge").

| Expert Model | Average Cos. Sim. | Winner Cos. Sim. | Judge #1 | Judge #2 |
|---|---|---|---|---|
| LLaMA 3.2 base model | 0.818 | 13.97% | 8.39% | 26.88% |
| LLaMA-TOMMI-1.0 | **0.879** | **86.02**% | **43.44**% | **43.23**% |
| Both models | - | - | 2.80% | 9.03% |
| Neither model | - | - | 45.38% | 20.86% |

Table 2: Results of the human evaluation user study comparing the LLaMA 3.2 base model with the fine-tuned LLaMA-TOMMI-1.0 version. Users could blindly select the LLaMA-3.2-11B base model as preferred, the LLaMA-TOMMI-1.0 model response as preferred, "both models" if responses were equally preferred, or "neither model" if both responses were poor. When considered as a stand-alone LLM response ("Expert Model Only / Advanced User"), we see a significant preference towards the LLAMA-TOMMI-1.0 model versus the LLaMA-3.2-11B base model. This preference is diminished when the combined platform is used, with a significant uptick in equally good responses from both models. However, a final (also blind) review by the FEM instructor reveals notably better course alignment when LLaMA-TOMMI-1.0 is used as a part of the platform response.

| Expert Model | Expert Model Only/ Advanced User | Platform / Advanced User | Platform: Course Alignment/ Instructor |
|---|---|---|---|
| LLaMA 3.2 base model | 25% | 30% | 28% |
| LLaMA-TOMMI-1.0 | **47**% | **34**% | **55**% |
| Both models | 16% | 32% | 10% |
| Neither model | 12% | 4% | 7% |

the maximum number of new tokens. For the synthesis model, the default is DeepSeek-R1-0528-Qwen3-8B. This choice is intentional, as our system prompt is carefully optimized for this reasoning model's specific behaviors. Alternate synthesis models include GPT-4o-mini and GPT-4.1-mini. Users can also adjust the synthesis output by changing its temperature and Top P.

## 4 Experimental Results

### 4.1 Automated & Human Evaluation of Expert Model

To measure the direct effect of fine-tuning, we first compared the performance of the fine-tuned expert model LLaMA-TOMMI-1.0 and its base model in a standalone setting on a test set of 465 question-answer pairs.

Automated evaluations are presented in Table 1, showing that LLaMA-TOMMI-1.0 consistently outperformed the base model. For cosine similarity, its average score was higher at 0.879 (compared to 0.818), and its response was closer to the ground truth in 86.02% of cases. Using a GPT-4o LLM-as-a-Judge, LLaMA-TOMMI-1.0 was again the preferred model for both Judge #1 (winning 43.44% vs. the base model's 8.39%) and Judge #2 (winning 43.23% vs. the base model's 26.88%).

These findings were consistent with human evaluations of the raw model outputs (Table 2, "Expert Model only" column): For this, we conducted a study involving four advanced users familiar with the course content, who evaluated model responses to real student questions using the AI-U framework. For each question, users made a blind preference judgment, selecting either the LLaMA 3.2 base model, the fine-tuned LLaMA-TOMMI-1.0 model, "both models" if responses were equally preferred, or "neither model" if

Table 3: Results from a second human evaluation user study for the stand-alone "expert" models. Users blindly rated the response from LLaMA-TOMMI-1.0, GPT5, and DeepSeek-R1-0528-Qwen3-8B for 80 questions and their corresponding answers. All three models were provided the same system instructions. Each stand-alone model was given a point if the response was deemed "sufficient". While these results show the larger reasoning models outperforming TOMMI, we note that the sophistication of the larger reasoning models renders this an expected outcome. This motivated a final (also blind) review by the FEM instructor, which revealed notably better course alignment when LLaMA-TOMMI-1.0 is used as a part of the platform response. See Table 4.

| Expert Model | Model only / Advanced User |
|---|---|
| GPT-5 | 92.5% |
| DeepSeek-R1-0528-Qwen3-8B | 82.5% |
| LLaMA-TOMMI-1.0 | 52.0% |

Table 4: Advanced users and the FEM course instructor weighed the strengths of GPT-5 (mathematical sophistication, broad coverage of advanced computational techniques) against the characteristics of the LLaMA-TOMMI-1.0 platform (alignment with the technical style, specific content and references to the FEM course). While reviewers were blinded to the identity of the two models, this was undone by the sharp difference in response style and content. We acknowledge that biases were probably introduced thereby.

| | Advanced User | Course Alignment / Instructor |
|---|---|---|
| OpenAI GPT-5 | 32.50% | 17.50% |
| LLaMA-TOMMI-1.0 Platform | 46.25% | 37.50% |
| Both | 18.75% | 41.25% |
| Neither | 2.50% | 3.75% |

both were responses were unsatisfactory. The advanced users preferred LLaMA-TOMMI-1.0 in 47% of cases and the base model in 25%. They found both answers to perform equally well 16% of the time and neither to be acceptable in 12% of cases.

As a final evaluation of the stand-alone expert model, we explore the performance of three models serving as "expert": GPT-5, DeepSeek-R1-0528-Qwen3-8B, and LLaMA-TOMMI-1.0 (Table 3). Each stand-alone model was given a point if the response was deemed "sufficient"—a combination of technical accuracy and completeness. While these results highlight that the larger reasoning models outperform TOMMI on a standalone basis, this conclusion is perhaps not unexpected, given their sophistication.

Table 5: Advanced users and the FEM course instructor weighed the strengths of the AI-U platform using either GPT-5 or LLaMA-TOMMI-1.0 platform as the expert agent. While reviewers were blinded to the identity of the two models, this was undone by the sharp difference in response style and content. We acknowledge that biases were probably introduced thereby.

| | Advanced User | Course Alignment / Instructor |
|---|---|---|
| Platform with OpenAI GPT-5 as expert | 11.25% | 10.00% |
| Platform with LLaMA-TOMMI-1.0 as expert | 15.00% | 73.75% |
| Both | 70.00% | 10.00% |
| Neither | 3.75% | 6.25% |

## 4.2 Human Evaluations of the Integrated Platform

Next, we assessed the practical performance of the models when integrated into our complete Platform (Figure 1). This human evaluation revealed a more nuanced picture (Table 2, "Platform / Advanced User"). A key feature of the platform's synthesis workflow is within its system prompt instruction (Technical Appendix, Section B.7 "Synthesis model")—it must evaluate the expert model's answer. If the answer is strong, it is used as a foundation and enriched with retrieved context. Conversely, if it is weak or inaccurate, the synthesis model must pivot to rely more heavily on the retrieved material to construct a new answer.

This compensation mechanism explains the results from advanced users, shown in Table 2 (Platform/Advanced User). The preference between models narrowed: LLaMA-TOMMI-1.0 was chosen 34% of the time and the base model 30%. The rate of "equally good" responses increased to 32%, while only 4% of responses were deemed unacceptable. This suggests that when the base model provided a weaker initial answer, the synthesis model successfully compensated by relying on the retrieved context, raising the quality of the final output to be comparable to that of the fine-tuned model.

However, the evaluation by the course instructor assessing correctness, tone, and overall alignment with the course showed that the quality of the initial expert model response still matters (Table 2, Platform: Course Alignment/Instructor). In this evaluation, the platform output using fine-tuned expert model LLaMA-TOMMI-1.0 was preferred in 55% of cases, versus 28% for the base model; both outputs were deemed acceptable 10% of the time, and neither was acceptable in the remaining 7%. This suggests that while the reasoning synthesis pipeline can improve weaker answers, starting with the fine-tuned response yields a superior outcome in most cases. It underscores the ongoing value of fine-tuning open-source LLMs for specialized educational applications.

Our comparison of OpenAI's GPT-5 against LLaMA-TOMMI-1.0 with the platform led to results with nuances and caveats (Table 4). Unsurprisingly, GPT-5 produced responses with high mathematical sophistication that exceeds the scope of the entry-level graduate FEM course on which LLaMA-TOMMI-1.0 has been fine-tuned. While GPT-5's answers were scientifically correct, they used concepts and referred to mathematical and computational techniques that would not be encountered by students in the chosen FEM course (or indeed most other entry-level graduate FEM courses) and its pre-requisites. When faced with questions for which it lacks context, GPT-5 switched from "Chat" to "Thinking" mode and provided multiple approaches to the question, many of which were relevant, even if beyond the scientific scope of the course. Some of its responses also asked for further information that experts trained beyond the course would recognize to be relevant in the broader setting of research into computational science.

These "strengths" of GPT-5 were weighed against the "characteristics" of the LLaMA-TOMMI-1.0 platform in responding to the same set of questions. These characteristics were typically good alignment with the course content by context, technical style, and provision of direct references to instructional content in the textbook and video lectures. Importantly, these sharp differences between the models rendered the blinded responses and randomization of order irrelevant. The advanced reviewers and course instructor were able to pinpoint with 100% accuracy the identity of the model producing a given response. The bias thus introduced must be recognized as a strong caveat in reviewing the results in Table 4. The above-mentioned "strengths" of GPT-5 were weighed against the "characteristics" of the LLaMA-TOMMI-1.0 platform to judge which was more suited to learning the specific FEM course. The advanced users preferred the LLaMA-TOMMI-1.0 platform over GPT-5 by a margin of 46.25% to 32.5%. The instructor, looking for alignment with the course's style, specific content and references preferred the LLaMA-TOMMI-1.0 platform over GPT-5 by a margin of 37.5% to 17.5%, while finding the "characteristics" of the former and the "strengths" of the latter to be in balance on 41.25% of the queries. We re-emphasize that biases entered these evaluations against GPT-5 only, because the models' identities were apparent from the style and content of their responses.

Similar trends are observed by comparing the AI-U platform with either GPT-5 or LLaMA-TOMMI-1.0 as the expert model 5. Once again, advanced users overwhelmingly approved of the responses for both models (70.00%), highlighting the ability of the synthesis agent to present a coherent and technically accurate response. Again, the instructor preferred the LLaMA-TOMMI-1.0 platform version (73.75%) due to its alignment with the course's style.

Recognizing that four advanced users and one instructor constitute a small set of evaluators, we conducted Inter-Annotator Agreement studies, which showed that these evaluators applied the rating criteria consistently relative to one another, reflecting a stable consensus within the user group. See Technical Appendix E "Inter-Annotator Agreement Analysis".

## 5 Conclusion

In this work, we present AI University (AI-U), a versatile and flexible framework designed to deliver bespoke science course content. By fine-tuning an LLM and incorporating a RAG system, AI-U creates an interactive learning environment that mirrors a course's specific instructional style. By framing our work in the setting of a graduate-level FEM class as an example, we demonstrate the framework's ability to generate accurate and highly relevant responses to course content by learning from a diverse set of course materials such as lecture video transcripts, notes, assignments, and textbooks.

The LLaMA 3.2 11-billion-parameter model optimized with LoRA performs well after hyperparameter tuning, and we validate its performance by cosine similarity and LLM-as-a-judge evaluations, with further human evaluation of its functionality by advanced users for correctness and usefulness in learning, and by the course instructor for correctness, tone and alignment with the instructional style of the course. Our evaluations of the LLaMA-TOMMI-1.0 and base LLaMA-3.2-11B parameter models by cosine similarity and LLM-as-a-judge showed a clear preference for the fine-tuned LLaMA-TOMMI-1.0 model. Notably, the advanced users' evaluations on the RAG and synthesis pipeline with reasoning suggested that this platform narrowed the gaps in the LLaMA-3.2-11B parameter base model. However, the instructor's evaluations for correctness, tone and alignment with the course carried out on the RAG and synthesis pipeline with reasoning also pointed to a clear preference for LLaMA-TOMMI-1.0. While the human evaluation was by a small cohort of advanced users and the instructor, Inter-Annotator Agreement analysis indicated that these evaluators had a consistent internal model of response correctness. We therefore consider our results to present strong validation that the adaptation of open-source LLMs to course-specific content can be successful with the approaches advanced here. We have not performed a comparison with other LLMs fine-tuned for educational use. Our searches found no other model that has been fine-tuned to specific course content on the FEM. Large commercial models such as GPT-5 bring mathematical sophistication and broad coverage of computational techniques to answering queries on FEM. However, there is room for considering whether the advanced technical level of these responses are less suitable for entry-level graduate learning of FEM than those well-aligned with the technical style and specific content of a course, backed by references to the instructional material.

A key feature of AI-U is its web application prototype, which not only provides comprehensive responses, but also enhances response credibility and traceability by linking to relevant course material. The framework is designed to be dynamic, supporting the continuous updating of new lecture content through RAG, ensuring that it remains consistent with the evolution of the course throughout the teaching term. This assistant extends learning beyond the classroom and can support discussions on platforms like Canvas or Piazza, where students often seek assistance outside of scheduled class hours. When instructors or teaching assistants are unavailable, the AI assistant can provide timely and contextually relevant responses that are oriented to the course's instructional style when presented with student queries. This research marks an important advancement in embedding AI into higher education, providing a scalable solution with the potential to enhance teaching efficiency and student engagement. We note, however, that full-fledged user studies have not been conducted by either student cohorts or instructors. Such a study would be crucial in flagging subtle errors introduced to the learner's mental model of the subject and could be revealed by follow-up questions in the manner of a live tutor-learner environment. Within the current framework, it could be implemented on the synthesis platform. These aspects are elements of a larger undertaking that exceeds the scope of the current work. We envision AI-U as a foundational tool that can be widely applied across academic fields, ultimately contributing to the construction of an integrated AI-enhanced university education system.

Finally, we note that our framework has been presented in the setting of a class on Finite Element Methods—a subject that is central to training PhD and Masters students in engineering science. However, it could have potential as a template in a broader context: fine-tuning LLMs to research content in science. In this

regard, our use of textbook, class notes and video lecture content could be supplemented by the broader technical literature, recorded research talks and simulations in a multi-modal learning environment. RAG, reasoning and multi-agentic inferencing would play important roles. Such a project could be addressed in future work.

## Broader Impact Statement

**AI-Generated Responses:** AI-U should be used responsibly. Answers are generated using AI and, while thorough, may not always be 100% accurate. Please verify the information independently.

**Content Ownership:** All video content and lecture material referenced belong to their original creators. The textbook *The Finite Element Method: Linear Static and Dynamic Finite Element Analysis* was used with permission of the author. All other course material was used with permission of the content creators. We encourage users to view the original material on verified platforms to ensure authenticity and accuracy. To ensure privacy, no student data was processed as a part of the training data generation pipeline.

**Review of Methods:** Each query used for human evaluation of the models was developed by consensus of anonymized groups of students who had consented to the use of their questions for instructional development. These students were not subjects in the studies, which did not interact or intervene with the students, nor include any access to identifiable private information. Advanced users were drawn from the authors of this work. The course instructor consented to and was fully aware of the methodological details and goals of this work.

**Educational Fair Use:** This tool is intended solely for educational purposes and operates under the principles of fair use. It is not authorized for commercial applications.

**Data Availability:** All code, study materials and results, and Q/A pairs will be made available upon publication. A partial, anonymous repo is available at: `https://anonymous.4open.science/r/AI-U-2025-TMLR/`

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

# Appendix

## A  Sample expert model responses

We present as a demonstration one sample test case, highlighting both the base model and LLaMA-TOMMI-1.0 responses in comparison with the default label answer. Note how in this example, context-specific variables $c$ and $\bar{d}$ specified by the user are interpreted incorrectly by the base model but correctly by LLaMA-TOMMI-1.0, when compared with the Label/Default answer.

[User Query]: In the context of 3D elasticity, how are the degrees of freedom
    for each node represented in the global 'c' and 'd bar' vectors?
════════════════════════════════════════════════
[Base Model Response (truncated for brevity]: In the context of 3D elasticity,
    the degrees of freedom (DOFs) for each node are typically represented in
    the global 'c' and 'd' vectors as follows:

Let's consider a 3D elasticity problem with a mesh consisting of 'n' nodes,
    each with 3 DOFs (displacements in the x, y, and z directions). The global
     'c' vector represents the coefficients of the stiffness matrix, and the
    global 'd' vector represents the forces applied to the system.

The global 'c' vector can be partitioned into three sub−vectors, each
    representing the coefficients of the stiffness matrix for a particular DOF
    :
    $$c=[c\_x\ c\_y\ c\_z]$$
where $c\_x$, $c\_y$, and $c\_z$ are sub−vectors of size 3nx3n representing the
    coefficients of the stiffness matrix for the x, y, and z DOFs,
    respectively.

Similarly, the global 'd' vector can be partitioned into three sub−vectors,
    each representing the forces applied to the system for a particular DOF:
    $$d=[d\_x\ d\_y\ d\_z]$$
where $d\_x$, $d\_y$, and $d\_z$ are sub−vectors of size $3n$ representing the
    forces applied to the system for the x, y, and z DOFs, respectively.

In the global stiffness matrix, the coefficients can be represented as...
════════════════════════════════════════════════
[LLaMA−TOMMI−1.0 Response]: In the context of 3D elasticity, the degrees of
    freedom for each node are represented in the global 'c' and 'd bar'
    vectors as follows:

− The global 'c' vector, which is a row vector, contains the degrees of
    freedom corresponding to the spatial dimensions (1, 2, and 3) for each
    node. For a node with a total of $N\_{sd}$ spatial dimensions, the 'c'
    vector will have $N\_{sd}$ entries for that node.
− The global 'd bar' vector, which is a column vector, also contains the
    degrees of freedom corresponding to the spatial dimensions (1, 2, and 3)
    for each node. For a node with a total of $N\_{sd}$ spatial dimensions, the
     'd bar' vector will have $N\_{sd}$ entries for that node.

These vectors are constructed by considering the spatial dimensions for each
    node and are used to represent the degrees of freedom in the global system
     of equations.

═══════════════════════════════════════════════════

[Label/Default Answer]: In the context of 3D elasticity, the degrees of
    freedom for each node are represented in the global 'c' and 'd bar'
    vectors as follows:

— **Global 'c' Vector**: This vector is defined by following the global nodes.
    For each global node, there are three degrees of freedom corresponding to
    the three spatial dimensions. For example, for global node 1, the degrees
    of freedom would be represented as $c_{11}$, $c_{12}$, and $c_{13}$,
    corresponding to spatial dimensions one, two, and three, respectively.
    This pattern continues for each node, such that for a general node A, the
    degrees of freedom are $c_{A1}$, $c_{A2}$, and $c_{A3}$.

— **Global 'd bar' Vector**: Similarly, the 'd bar' vector is structured to
    represent the displacement degrees of freedom for each node in the
    respective coordinate directions. For example, for node 1, the degrees of
    freedom are represented as $d_{11}$, $d_{12}$, and $d_{13}$. For a general
    node A, the degrees of freedom are $d_{A1}$, $d_{A2}$, and $d_{A3}$,
    representing displacements in the directions one, two, and three,
    respectively.

The 'd bar' vector is typically larger than the 'c' vector because it includes
    all degrees of freedom before accounting for Dirichlet boundary
    conditions, which may reduce the number of active degrees of freedom in
    the 'c' vector.

## B   Prompts

### B.1   Textbook question generation

The following prompt is used to generate questions from the textbook and course transcripts.

    You are an AI assistant specialized in creating educational content for
        Finite Element Method (FEM).
    Generate comprehensive set of questions on topics related to FEM from the
        input text. **Only questions, no answer is needed.** Follow these
        guidelines:

    1. Questions:
    — Focus on fundamental concepts, theories, and general applications of FEM
        .
    — Ensure that the questions are relevant to the input text, and can be at
        least partially answered using the provided text.
    — Emphasize broad understanding rather than niche knowledge.
    — Questions can be of any length needed to fully express the concept being
        tested.
    — Complex questions involving multiple parts or mathematical derivations
        are encouraged.
    — Each question should have all the information needed such that it makes
        sense without referencing the input text.
    — Any variables that are used in the question must be defined in the
        question.
    — Provide enough information such that the question makes sense without
        referencing a specific chapter or section.

- Do not refer to the proof number in the question text when generating
  questions about a proof.
- Add a description of any proofs used when generating questions about
  proofs.

2. Coverage:
- For each question, include a "coverage" field.
- In this field, estimate the percentage of the possible answer that is
  covered by the input text.
- Use your judgment to assign a realistic percentage in integer form,
  considering the depth and specificity of the input text.

Note: Mathematical Notation:
- Use LaTeX formatting for mathematical expressions
- For inline equations, use single $ wrapper (e.g., "Calculate the strain
  energy $U = \\frac{1}{2}\\int_V \\sigma\\epsilon dV$")
- For display equations, use double $$ wrapper, e.g.:
    "Derive the stiffness matrix given the following stress-strain
        relationship:
    $$
    \\begin{{bmatrix}}
    \\sigma_{{xx}} \\\\ \\sigma_{{yy}} \\\\ \\tau_{{xy}}
    \\end{{bmatrix}} =
    \\begin{{bmatrix}}
    D_{{11}} & D_{{12}} & 0 \\\\
    D_{{21}} & D_{{22}} & 0 \\\\
    0 & 0 & D_{{33}}
    \\end{{bmatrix}}
    \\begin{{bmatrix}}
    \\epsilon_{{xx}} \\\\ \\epsilon_{{yy}} \\\\ \\gamma_{{xy}}
    \\end{{bmatrix}}
    $$"

Note: Your response format as JSON must adhere to the following structure:
[
{{
    "question": "What are the shape functions and their role in accuracy
        of approximations?",
    "coverage": 95
}},
{{
    "question": "How are boundary conditions imposed? Explain elimination
        approach.",
    "coverage": 70
}}
]
Do not include the word JSON at the start of the response.
Generate as many questions as needed to cover the input text, up to {k}
    diverse questions, with Coverage 30-100 percentage.

## B.2  Textbook answer generation

The following prompt is used to generate answers for the textbook and course transcript questions.

```
You are an AI teaching assistant for a Finite Element Method (FEM) course.
    Answer questions based EXCLUSIVELY on the provided context. If context is
    insufficient for a very accurate answer, respond with: Answer: "NOT ENOUGH
    INFO."

    If context is sufficient:

    1. Answer Guidelines:
    - Use only information from the context
    - Restrict your use of finite element method knowledge to what is provided
         in the context provided. Do not use additional background finite
         element method knowledge in generating the answer (you may use
         background knowledge from other areas).
    - Show step-by-step work for calculations
    - For multiple valid interpretations, provide separate answers

    2. Mathematical Notation:
    - Use $ for inline equations (e.g., $U = \\frac{{1}}{{2}} \\int_V \\sigma
         \\epsilon dV$)
    - Use $$ for display equations, especially matrices:
    $$
    \begin{{bmatrix}}
    \sigma_{{xx}} & \sigma_{{xy}} \\
    \sigma_{{yx}} & \sigma_{{yy}}
    \end{{bmatrix}}
    $$

    Note: Focus on FEM fundamentals, theories, and applications as presented
        in the context.
    """

    user_prompt = f"""
    Context:
    {context}

    Question:
    {question}

    Answer (based EXCLUSIVELY on the above context):
```

## B.3   Fine-tuning system prompt

The following system prompt is provided to the LLM during fine-tuning of LLaMA-TOMMI-1.0.

```
You are an AI professor for a Finite Element Method (FEM) course. You are
    asked a question by a student and return an appropriate answer based on
    course material. Your response focuses on FEM fundamentals, theories, and
    applications as presented in the course. Use standard latex notation when
    replying with mathematical notation.
```

## B.4   Coding question-answer generation prompt 1

The following prompt is one of three that was used to generate questions and answers from previous course coding assignments.

You are an expert in finite element methods (FEM), the deal.II library, and C
  ++. You are tasked with creating detailed question−answer pairs for a
  coding assignment. The assignment description, along with the solution
  files ('main.cc' and 'fem.h'), is provided. Follow these detailed
  instructions to generate the Q&A pairs:

  1. **Functions as Answers:** Each answer must include the implementation
     of individual functions or classes from the code files.

  2. **Cover All Code Components:** Generate questions for every function,
     constructor, destructor, and class definition in both 'main.cc' and '
     fem.h'. Ensure that no code component is left out.

  3. **Detailed Question Context:** Each question must:
     − Include a **general problem statement** derived from the assignment
        description to provide a clear context.
     − Stand alone, without referencing the assignment, other questions, or
        answers, so that it makes sense independently.
     − Clearly ask for the specific function, constructor, destructor, or
        class related to the problem context.
     − Mention that the answer can use the open source library dealii

  4. **Variety in Questions:** In addition to asking for individual
     functions:
     − Include questions that require the entire class implementation as an
        answer (e.g., the 'FEM' class).
     − Include a question asking for the names of all functions required to
        solve the assignment.

  5. **Formatting:** Use the following format for the Q&A pairs. Make sure
     not to number them:
     ' ' '
     Q: <Insert detailed question here>
     A: <Insert complete function/class implementation here>
     ' ' '

  6. **Descriptive Questions:** The questions should be long enough and
     verbose so that they are standalone and cover all the descriptive
     background from the original assignment without refering to the
     assignment.

  7. **Example Question for Context:** Use the style below as a reference
     for detailing each question:
     − Example Q:
       Consider the following differential equation of elastostatics, in
          strong form: \\ \\
       Find $u$ satisfying
       \begin{displaymath}
       (E\,A\, u_{,x})_{,x} + f\,A = 0,\quad \mbox{in}\; (0,L),
       \end{displaymath}
       \noindent for the following sets of boundary conditions and forcing
       function ($\bar{f}$ and $\hat{f}$ are constants):
       \begin{itemize}
        \setlength{\itemsep}{0pt}
        \item[(\romannumeral 1)]$u(0) = g_1$, $u(L) = g_2$, $f = \bar{f} x$,

```
\item [(\romannumeral 2)] $u(0) = g_1$, $EAu_{,x} = h$ at $x = L$, $f
    = \bar{f} x$,
\end{itemize}
```

When writing a one−dimensional finite element code in C++ using the deal.II FEM library framework to solve the given problem, what will the class constructor look like?

− Example A:
   Here is the class constructor to solve this problem:
   ```
   ' ' '
   template <int dim>
   FEM<dim>::FEM (unsigned int order, unsigned int problem)
   : fe (FE_Q<dim>(QIterated<1>(QTrapez<1>(),order)), dim),
     dof_handler (triangulation)
   {
     basisFunctionOrder = order;
     prob = problem;
     for (unsigned int i=0; i<dim; ++i){
       nodal_solution_names.push_back("u");
       nodal_data_component_interpretation.push_back(
           DataComponentInterpretation::component_is_part_of_vector);
     }
   }
   ```
   Here are the files related to the coding assignment:

1. Assignment Description:
{assignment_description}

2. Contents of main.cc:
{main_code}

3. Contents of fem.h:
{fem_code}

### B.5 Coding question-answer generation prompt 2

The following prompt is one of three that was used to generate questions and answers from previous course coding assignments.

You are an expert in finite element methods (FEM), the deal.II library, and C++. You are tasked with creating detailed question−answer pairs for a coding assignment. The assignment description and the coding template file, along with the solution files ('main.cc' and 'fem.h'), are provided. Follow these detailed instructions to generate the Q&A pairs:

1. **Test on identical material/information as the provided assignment template:** Question Answer pairs must be based on what the coding assignment is targeting the student to understand. The student is expected to use the template coding files and fill them to get the solution coding files. Match the differences between the coding template files and the coding solution and base your question−answers on this. Essentially the QA pairs generated should quiz the student on

the identical material tested by the coding assignment and the
    provide coding template.

2. **Detailed Question Context:** Each question must:
   − Include a **general problem statement** derived from the assignment
       description to provide a clear context.
   − Stand alone, without referencing the assignment, other questions, or
       answers, so that it makes sense independently.
   − Clearly ask for the specific function, constructor, destructor, or
       class related to the problem context as in the previous point.
   − Mention that the answer can use the open source library dealii
   − The questions should be long enough and verbose so that they are
       standalone and cover all the descriptive background from the
       original assignment without refering to the assignment.
   − If the assignment asks for something particular to be implemented
       such as the boundary condition (pde variables, mesh variables etc),
        the question should list the boundary conditions to be implemented
        .

3. **Generate as many questions:** Cover all the assignment problem
    specific implementations in the code even if they are already provided
     in the template files.

4. **Formatting:** Use the following format for the Q&A pairs. Make sure
    not to number them:
    ```
    Q: <Insert detailed question here>
    A: <Insert function/class implementation here>
    ```

    Here are the files related to the coding assignment:

1. Assignment Description:
{assignment_description}

2. Contents of Template main.cc:
{templateMain}

3. Contents of template fem.h:
{templateFEM}

4. Contents of solution main.cc:
{main_code}

5. Contents of solution fem.h:
{fem_code}

## B.6 Coding question-answer generation prompt 3

The following prompt is one of three that was used to generate questions and answers from previous course
coding assignments.

You are an expert in finite element methods (FEM), the fenics library, and
    python. You are tasked with creating detailed question−answer pairs for a
    coding assignment. The assignment description, along with the solution

file ('fem.h'), is provided. Follow these detailed instructions to generate the Q&A pairs:

1. **Answers Based On Code:** Answers should be based on code implementation.

2. **Cover All Code Components:** Generate as many questions using 'fem.h' ensuring no code component is left out. More the questions, the better. It is ok if some questions are repeated/have some overlap.

3. **Detailed Question Context:** Each question must:
   – Include a **general problem statement** derived from the assignment description to provide a clear context.
   – Stand alone, without referencing the assignment, other questions, or answers, so that it makes sense independently.
   – Clearly ask for the specific code implementation related to the problem context.
   – Mention that the answer should be based on open source finite element library fenics
   – The questions should be long enough and verbose so that they are standalone and cover all the descriptive background from the original assignment without refering to the assignment.
   – If the assignment asks for something particular to be implemented such as the boundary condition (pde variables, mesh variables etc), the question should list the boundary conditions to be implemented.
   – Make sure to not refer to the assignment.

4. **Formatting:** Use the following format for the Q&A pairs. Make sure not to number them:
   ```
   Q: <Insert detailed question here>
   A: <Insert complete function/class implementation here>
   ```
   Here are the files related to the coding assignment:

1. Assignment Description:
{assignmentDescription}

2. Contents of fem.h:
{femCode}

## B.7 Synthesis model

The following is the system prompt used by the synthesis model.

You are an AI teaching assistant for a {subject_matter} course. Your task is to synthesize a final, high-quality answer to the student's **Question** by intelligently integrating two sources: a preliminary **Direct Answer** and the official **Retrieved Context** from the course materials.
By synthesizing we mean that your final answer must always be grounded **exclusively** in the provided **Direct Answer** and **Retrieved

Context**. Therefore, never use any external knowledge including your existing knowledge.
IMPORTANT INITIAL CHECK: Analyze the provided **Question**, **Direct Answer**, and **Retrieved Context**.
If the **Direct Answer** AND the **Retrieved Context** together lack sufficient information to answer the **Question**, respond EXACTLY as follows and then STOP:
"NOT_ENOUGH_INFO The provided context doesn't contain enough information to fully answer this question. You may want to increase the number of relevant context passages or adjust the options and try again."
Else continue with the remaining guidelines.
Guidelines:
1. Your primary synthesizing goal is to use the **Retrieved Context** to validate, improve, and expand upon the **Direct Answer**.
    a. If the **Direct Answer** is accurate and relevant, use it as the foundation for your response. Your task is then to enrich it by weaving in specific details, examples, and citations from the **Retrieved Context** to create a more comprehensive and well−supported answer.
    b. If the **Direct Answer** is poor, inaccurate, or irrelevant, you should rely more heavily on the **Retrieved Context** to construct the correct answer from the ground up.
2. Referencing:
    a. Always cite your sources by referencing the video number and the given time in brackets and **bold** (e.g., [**Video 3, time 03:14**]) after each piece of information you use in your answer.
    b. You may cite multiple references if they discuss the same content ( e.g., [**Video 3, time 03:14; Video 1, time 12:04**]). However, try to reference them separately if they cover different aspects of the answer.
3. Style and Formatting:
    a. Provide the answer in markdown format. Any latex formating should be converted to an equivalent markdown format.
    b. Do not use any titles, sections, or subsections. Use mainly paragraphs. Bold text, items, and bullet points if it helps.
    c. Symbols and equations within the text MUST be placed between $ and $, e.g., $x=0$ is the min of $\sigma(x)=x^2$.
    d. For equations between paragraphs, use \\n\\n$ and $\\n\\n. For example, in the following equation: \\n\\n$ E = mc^2 $\\n\\n, note $c$ as the speed of light. Remove any equation number/tags in the raw data.
4. Use technical language appropriate for a {subject_matter} course, but be prepared to explain complex terms if asked.
5. If the question involves calculations, show your work step−by−step, citing the relevant formulas or methods from the context.

## B.8 LLM-as-a-judge

The following prompts are used as a part of the LLM-as-a-Judge evaluation.

Judge 1:

Evaluate which model response better aligns with the professor's reference answer to a question ONLY based on the following three key dimensions :

1. Lexical matching: Does the response use the same key terms, phrases, and specific wording as the professor's answer?
2. Structural similarity: Does the response follow the same order and organization of ideas as the professor's answer?
3. Example consistency: Does the response use the same specific examples as the professor's answer?

Return your decision as a JSON object:
— "winner": "model 1", "model 2", "neither" (if both significantly diverge), or "both" (if equally aligned)
— "justification": A brief explanation of your choice based on the three dimensions above

Question: {question}
Professor's Answer (Reference): {prof_ans}
Model 1 Response: {base_model}
Model 2 Response: {fine_tuned}

Output only the JSON.

Judge 2:

Evaluate which model response better aligns with the professor's reference answer to a question ONLY based on the following three key dimensions:

1. **Content Accuracy**: Does the response convey correct information without factual errors or misconceptions?
2. **Conceptual Alignment**: Does the response reflect the professor's key ideas and reasoning, even if phrased differently or in a different order?
3. **Completeness**: Does the response fully address all parts of the question that the professor addressed?

Return your decision as a JSON object:
— "winner": "model 1", "model 2", "neither" (if both responses contain major issues), or "both" (if equally strong)
— "justification": A brief explanation of your choice based on the three dimensions above

Question: {question}
Professor's Answer (Reference): {prof_ans}
Model 1 Response: {base_model}
Model 2 Response: {fine_tuned}

Output only the JSON.

## C  Question/Answer pair evaluation

To establish confidence in the training data, 200 question and answer pairs were randomly selected and reviewed by the authors who are experts in the field. Of these, 5 pairs were not related to course material and deemed inappropriate for the expert evaluation. An additional 14 were identified as having poor quality and potentially inappropriate for training. Of these 14, 5 were poorly posed questions that returned a response

of "Not Enough Information" from the generation pipeline. (Note that, while these were not relevant to the human expert reviews, they *do* aid in training the model to appropriately identify poorly phrased questions). The remaining 9 pairs were flagged as having sufficient technical shortcomings that would disqualify them for training. This relatively low error rate of 4.5% is well within the standard operational tolerance for state-of-the-art model training and is on par with widely accepted datasets Northcutt et al. (2021). Furthermore, we note that the expert model is but one part of the overall framework. As highlighted by the user studies, we have noted significant improvement of the responses when combined with the synthesis model.

## D    Platform evaluation

Please see *Supplementary Material* for screenshots of the output from the platform.

## E    Inter-Annotator Agreement Analysis

To address the limited number of evaluators ($N = 5$ from four advanced users and one instructor) and validate the rigor of our statistical claims, we conducted a post-hoc Inter-Annotator Agreement (IAA) analysis. We collected responses from the 4 advanced users and the instructor across a set of 80 identical questions used for the study in Section 4.2 of the responses of LLaMA-TOMMI-1.0 and GPT-5 on the integrated platform, to evaluate the consistency of their ratings. The rating scale consisted of four categories: '1' (Model 1 preferred), '2' (Model 2 preferred), 'eg'(both models equally good), and 'eb'(both models equally bad). The evaluators were blind to the identities of Models 1 and 2.

### E.1    Consistency Among Advanced Users

First, we examined the agreement levels exclusively among the four advanced users to determine if the participants shared a consistent mental model when evaluating the system's responses. From Table 6, we have:

1. Pairwise Agreement: The pairwise Cohen's Kappa scores(Cohen, 1960) ranged from 0.34 to 0.69, with a mean of 0.54.

2. Group Consensus: The Fleiss' Kappa(Fleiss, 1971) for all four advanced users was 0.529, which is interpreted as Moderate Agreement.

We re-emphasize that the platform's improvement of the models' responses with RAG and reasoning resulted in more subtle differences between the responses of the models, and a large number of 'eg' evaluations by the Advanced Users. Even still, the observed level of agreement indicates that despite the more subtle nature of the task, the Advanced Users applied the rating criteria consistently relative to one another. This suggests that the data collected from the user study reflects a stable consensus within the user group.

### E.2    Comparison with the Instructor

We also compared the ratings of each Advanced User against the Instructor's ratings to assess alignment with an expert baseline.

From table 7, The mean Cohen's Kappa between the Instructor and the Advanced Users was 0.11 (Slight Agreement). Detailed analysis reveals a systematic divergence: many QA pairs rated as 'eg' by the advanced users were assigned preferred ratings ('1' or '2') by the instructor. This discrepancy stems from the instructor's unique role as the course creator and developer of the training corpus. Consequently, the instructor applied significantly stricter criteria, possessing the domain expertise to discern subtle nuances in the QA pairs that determine their true suitability and alignment with the specific course curriculum.

Given this distinction, we reported the instructor's evaluation independently rather than aggregating it with the advanced user data (acknowledging that the study was limited to a single domain expert). Furthermore, the high uniformity of the Cohen's Kappa scores observed across all four Instructor-vs-User pairs serves as

additional evidence of the Advanced Users' internal consistency. It indicates that the users shared a stable evaluation model, even if that model was systematically more lenient than the Instructor's gold standard.

Table 6: **Inter-Annotator Agreement among Advanced Users.** The moderate to substantial agreement (Mean Kappa = 0.536, Fleiss' Kappa = 0.529) indicates a consistent mental model and high reliability within the user group.

| Comparison Pair | Cohen's Kappa | Agreement Rate | Interpretation |
|---|---|---|---|
| User 1 vs. User 2 | 0.539 | 76.25% | **Moderate** |
| User 1 vs. User 3 | 0.442 | 71.25% | **Moderate** |
| User 1 vs. User 4 | 0.341 | 62.50% | Fair |
| User 2 vs. User 3 | 0.688 | 85.00% | **Substantial** |
| User 2 vs. User 4 | 0.558 | 76.25% | **Moderate** |
| User 3 vs. User 4 | 0.646 | 81.25% | **Substantial** |
| **Mean Pairwise** | **0.536** | **75.42%** | **Moderate** |
| **Fleiss' Kappa (Global)** | **0.529** | — | **Moderate** |

Table 7: **Alignment between Instructor (Expert) and Advanced Users.** While exact ratings diverge due to different expert perspectives (Mean Kappa = 0.109), the agreement rate remains consistent across all user-instructor pairs ($\sim$43%).

| Comparison Pair | Cohen's Kappa | Agreement Rate | Interpretation |
|---|---|---|---|
| Instructor vs. User 1 | 0.110 | 42.50% | Slight |
| Instructor vs. User 2 | 0.074 | 41.25% | Slight |
| Instructor vs. User 3 | 0.149 | 46.25% | Slight |
| Instructor vs. User 4 | 0.104 | 42.50% | Slight |
| Average Alignment | 0.109 | 43.13% | Slight |

