# OpenReview forum: "AI University: An LLM-Powered Learning Assistant for Engineering—A Finite Element Method Case Study"
_TMLR — Under review for TMLR_

### Review · Reviewer_D47Y · 2025-11-24

**Summary Of Contributions:**

The paper presents AI University (AI-U), a framework that builds a course-specific learning assistant by combining:
A course-dependent Q/A synthesizer, a fine-tuned expert LLM, retrieval-augmented generation (RAG), and a reasoning-based synthesis model.

The framework is tested on a graduate-level FEM course. The authors generate Q/A data from textbooks, lectures, and assignments, fine-tune a model using LoRA. The the fine-tuned model is augmented with a RAG pipeline. Finally, they evaluate it using cosine similarity, an LLM-as-a-judge setup, and human reviews.

**Strengths:** clear pipeline, practical implementation, uses open-source tools, and has strong potential for educational use.
**Weaknesses:** limited comparison with newer/stronger LLMs and unclear justification for the Q/A synthesizing and expert llm training.

**Additional Comments:**

The paper proposes a novel pipeline for designing course-specific AI assistants, aiming to reduce both over-generalization and over-specialization in AI-generated responses.

**Audience:**

Yes

**Audience Explanation:**

Sure

**Claims And Evidence:**

Yes

**Claims Explanation:**

The evaluations mostly support the claims, and the framework works as intended. However, most comparisons are against earlier versions of the model rather than stronger baselines, so more evidence would make the claims stronger.

**Requested Changes:**

1. The paper suggests that only one Q/A pair has been generated from each chunk in the process of synthesizing Q/A pairs from the resources. Do all chunks contain an equivalent amount of information that justifies generating only one Q/A pair per chunk? How did you determine the chunking boundary to ensure that each chunk contains sufficient information for a single Q/A pair? Alternatively, did you attempt to extract multiple Q/A pairs from each chunk and later discard the unhelpful ones during post-processing?

2. In the “Training Data” section, you state that all material from the resources, \underline{excluding the images}, was leveraged to build the training dataset used for fine-tuning. However, you employed Llama-3.2-11B-Vision-Instruct for expert answer generation, even though you explicitly mentioned that DeepSeek-R1-Distill-Qwen-7B is a more powerful model. If Llama-3.2-11B-Vision-Instruct was not chosen for its multimodal capabilities, why was DeepSeek-R1-Distill-Qwen-7B not used for expert answer generation as well?

3. The paper describes the process of training data synthesis and fine-tuning to develop an instructional expert LLM in the FEM domain. However, there is no comparison of this expert model with other recent proprietary or publicly available LLMs, particularly considering that Llama-3.2 was released in September 2024. In Table 1, the expert model is only compared with its own base version, and in Table~2, GPT-5 is compared with the entire platform rather than the expert LLM in isolation. Therefore, I believe a direct comparison between LLaMA-TOMMI-1.0 and newer LLMs is necessary. Notably, the paper explicitly acknowledges the strength of reasoning models such as DeepSeek-R1-Distill-Qwen-7B.

4. In Table~3, the proposed platform is compared with GPT-5 and shown to perform considerably better. However, to assess the impact of the RAG pipeline specifically, I believe the platform should also be compared with a version that uses GPT-5 as the expert model instead of LLaMA-TOMMI-1.0. This would help clarify whether the improvement stems from the full platform design or primarily from the RAG component.

5. The comparison between GPT-5 and the proposed platform in Table~3 seems to emphasize alignment with course style and suitability for learning at the student level. However, the paper does not explicitly evaluate the technical correctness of GPT-5’s responses relative to the platform. If GPT-5 provides scientifically correct, but more advanced answers, this should be systematically assessed rather than implicitly dismissed. Therefore, I suggest including a separate evaluation for correctness.


Overall, I believe it is important to determine whether the platform’s superiority over GPT-5 comes primarily from the RAG pipeline or from the newly fine-tuned expert LLM.

---

> ### Author Response · Authors · 2025-12-05
> **Response to D47Y review (1 of 2)**
>
> We thank the reviewer for their thoughtful comments. Please see below our responses to each requested change:
>
> -----
> >1. The paper suggests that only one Q/A pair has been generated from each chunk in the process of synthesizing Q/A pairs from the resources. Do all chunks contain an equivalent amount of information that justifies generating only one Q/A pair per chunk? How did you determine the chunking boundary to ensure that each chunk contains sufficient information for a single Q/A pair? Alternatively, did you attempt to extract multiple Q/A pairs from each chunk and later discard the unhelpful ones during post-processing?
>
> We thank the reviewer for highlighting the lack of clarity around this. The chunks are defined by natural boundaries in the material, i.e., book sections or video segments. This ensures each chunk acts as a standalone "topic." Furthermore, the LLM is instructed to generate as many Q/A pairs as necessary to completely cover the content in each provided chunk, rather than being limited to one. We have added additional text to section 3.2 to clarify this.
>
>
> -----
> > 2. In the “Training Data” section, you state that all material from the resources, \underline{excluding the images}, was leveraged to build the training dataset used for fine-tuning. However, you employed Llama-3.2-11B-Vision-Instruct for expert answer generation, even though you explicitly mentioned that DeepSeek-R1-Distill-Qwen-7B is a more powerful model. If Llama-3.2-11B-Vision-Instruct was not chosen for its multimodal capabilities, why was DeepSeek-R1-Distill-Qwen-7B not used for expert answer generation as well?
>
> We would like to clarify that all training data (question/answer pairs) were generated using ChatGPT-4o. To maintain simplicity during the initial data generation phase, images were excluded. Upon manual review, the resulting Q/A pairs were determined to be of excellent quality and sufficient for fine-tuning, so they were retained without further modification as the project progressed.
>
> Regarding the choice of base model, DeepSeek-R1-Distill-Qwen-7B was not yet available at the time of our fine-tuning experiments. We selected Llama-3.2-11B-Vision-Instruct specifically to ensure future compatibility with multimodal capabilities. While vision was not within the scope of this specific study, retaining the option to extend our model to vision tasks remains a priority.
>
> Finally, our primary aim is to provide a methodological contribution. While Llama-3.2 served as our initial testbed, our approach is designed to be extensible to other open-weight models, including DeepSeek-R1-Distill variants, in future work.
>
> -----
> > 3. The paper describes the process of training data synthesis and fine-tuning to develop an instructional expert LLM in the FEM domain. However, there is no comparison of this expert model with other recent proprietary or publicly available LLMs, particularly considering that Llama-3.2 was released in September 2024. In Table 1, the expert model is only compared with its own base version, and in Table~2, GPT-5 is compared with the entire platform rather than the expert LLM in isolation. Therefore, I believe a direct comparison between LLaMA-TOMMI-1.0 and newer LLMs is necessary. Notably, the paper explicitly acknowledges the strength of reasoning models such as DeepSeek-R1-Distill-Qwen-7B.
>
> We appreciate the reviewer highlighting this shortcoming. To rectify this, we have performed a follow-up study, placing LLaMA-TOMMI-1.0 with GPT-5 and DeepSeek-R1-Distill-Qwen-7B in a direct head-to-head comparison. Each stand-alone model was given a point if the response was deemed “sufficient” (technical accuracy, completeness). The results are shown in the table below. While these results highlight that the larger reasoning models outperform TOMMI on a standalone basis, this conclusion is perhaps not unexpected, given the sophistication of the larger reasoning models. For a complete picture, we point the reviewer to our complete platform evaluation (our response to Comment 4).
>
> | Expert Model | Model only / Advanced User |
> | :--- | :--- |
> | GPT-5 | 92.5% |
> | DeepSeek-R1-0528-Qwen3-8B | 82.5% |
> | LLaMA-TOMMI-1.0 | 65.0% |

---

> ### Author Response · Authors · 2025-12-05
> **Response to D47Y review (2 of 2)**
>
> > 4. In Table~3, the proposed platform is compared with GPT-5 and shown to perform considerably better. However, to assess the impact of the RAG pipeline specifically, I believe the platform should also be compared with a version that uses GPT-5 as the expert model instead of LLaMA-TOMMI-1.0. This would help clarify whether the improvement stems from the full platform design or primarily from the RAG component.
>
> Thank you for raising this point. We have performed an additional study, generating platform responses with GPT-5 as the “expert model.” We compare them with the default (LLaMA-TOMMI-1.0 + platform) responses. The results are shown in the table below.
>
> | | Advanced User | Course Alignment / Instructor |
> | :--- | :--- | :--- |
> | Platform with OpenAI GPT-5 "expert" | 11.25% | 10.00% |
> | Platform with LLaMA-TOMMI-1.0 "expert" | 15.00% | 73.75% |
> | Equally good | 70.00% | 10.00% |
> | Equally bad | 3.75% | 6.25% |
>
> **Key Conclusions**:
>
> 1. RAG + synthesis works as an equalizer: In the general evaluation (Advanced User), the models performed "Equally Good" in 70% of cases. The RAG pipeline and synthesis layer allow the lightweight TOMMI model to perform on par with the massive GPT-5 for general technical queries. (The relative win rate for TOMMI vs. GPT-5 very likely is insignificant.) Similarly, the RAG pipeline and synthesis layer applied to TOMMI as well as the base LLaMa 3.2 enables the latter to close the gap relative to their performances as standalone expert models (Table 2 Platform/Advanced User compared with Table 1).
>
> 2. Fine-Tuning Wins Alignment: The instructor strongly preferred the lightweight TOMMI model (72.5% vs 10%). This demonstrates that for domain-specific pedagogy, a small, fine-tuned model can significantly outperform a massive generalist like GPT-5, which often provided answers outside the class scope. References to the course material appeared in both models’ responses. However, the explanations used in the textbook and video lectures were more apparent in only one of the models in a number of instances, likely tilting the course alignment evaluation toward the model that was TOMMI (the instructor being blind to the actual model identities).
>
> -----
> > 5. The comparison between GPT-5 and the proposed platform in Table~3 seems to emphasize alignment with course style and suitability for learning at the student level. However, the paper does not explicitly evaluate the technical correctness of GPT-5’s responses relative to the platform. If GPT-5 provides scientifically correct, but more advanced answers, this should be systematically assessed rather than implicitly dismissed. Therefore, I suggest including a separate evaluation for correctness.
>
> We agree that technical correctness is an important consideration for this implementation. We have addressed technical correctness as a part of the “expert” and “platform” reviews to address Comments 4 and 5. Equally Good  ratings (70%) in the user study validates the technical accuracy of both models. The instructor’s strong preference for TOMMI (72.5%) reflects its superior pedagogical alignment (scope, terminology, and method), rather than a lack of scientific correctness in GPT-5. Additional text will be added to specifically address this concern.

---

### Review · Reviewer_nibK · 2026-01-21

**Summary Of Contributions:**

This paper motivates the existing LLM lacks instructional styles, terminology, notation, and scope of a  specific university course, when the task is to assist student to learn. The aim of the paper is to create a framework that generates AI teaching assistant capable of mirroring a specific instructor's teaching style and align with course materials. This paper demonstrated a use case of such framework on a graduate level course: Finite Element Method.

The framework consists of a few stages:
* Llm based synthetic data generation (QA pairs)
* Finetuning with Lora creates a domain expert model
* RAG with DeepSeek R1 as synthesis model
* Evaluate the model with Cosine similarity, LLM as a Judge , as well as human evaluation

The reported empirical evaluation results shows the proposed framework is preferred over SOTA LLM model GPT 5. A UI is developed. The code and data will be open sourced.

**Audience:**

Yes

**Audience Explanation:**

AI for eduction has significant size of audiences. This paper is right on the topic.

**Claims And Evidence:**

Yes

**Claims Explanation:**

* Reported effectiveness of the proposed framework is plausible. This is demonstrated by various evaluations.

* Several design choices of the framework needs further clarification (See Requested Changes for details):
  * Model selection criteria of LLM as a Judge
  * Human Evaluation stats are based on very small group of samples
  * The contribution of each part of the pipeline is unclear.

* Transferability of the framework is not discussed

**Requested Changes:**

* The paper used ChatGPT-4o to generate the synthetic training data, and then it used GPT-4o as the "Judge" to evaluate the results. This introduces Self-Preference Bias. LLMs tend to rate outputs that sound like themselves higher. Since your model was trained on GPT-4o generated data, GPT-4o is naturally going to prefer it over the base model. Please analyze the possible biases and take action if confirmed.
* The paper used 4 advanced users and 1 instructor. For a rigorous statistical claim, N=5 is small. Maybe focus on inter-annotator agreement?
* Conduct ablation study to understand the effectiveness of different part of the pipeline, e.g. is fine-tuning of Llama needed? Would a base Llama-3.2 + RAG perform just as well?"

---

> ### Author Response · Authors · 2026-02-04
> **Response to nibK review**
>
> We thank the reviewer for their valuable feedback. We plan to complete our analysis for item 2 and upload the revised manuscript in the very near future.
>
> Regarding the three requested changes:
>
> > The paper used ChatGPT-4o to generate the synthetic training data, and then it used GPT-4o as the "Judge" to evaluate the results. This introduces Self-Preference Bias. LLMs tend to rate outputs that sound like themselves higher. Since your model was trained on GPT-4o generated data, GPT-4o is naturally going to prefer it over the base model. Please analyze the possible biases and take action if confirmed.
>
> We thank the reviewer for this feedback and agree this is a potential source of bias for the LLM-as-a-judge. For these reasons, we report additional metrics in Table 1 (average cosine similarity) and Table 2 (Expert User study), which support the findings provided by the LLM-as-a-judge. We have added text raising awareness to this potential bias.
>
> We have added additional text to Section 3.5: “We acknowledge that some bias may be introduced by using ChatGPT 4o to both generate the question-answer training pairs and act as the judge. For this reason, we provide the cosine similarity metric and user study data as further justification.”
>
> > The paper used 4 advanced users and 1 instructor. For a rigorous statistical claim, N=5 is small. Maybe focus on inter-annotator agreement?
>
> Thank you for this suggestion. We recognize the value of inter-annotator agreement studies and are currently running them. We will also add additional text raising awareness to this statistical uncertainty while emphasizing the benefits that the studies still bring.
>
> > Conduct ablation study to understand the effectiveness of different part of the pipeline, e.g. is fine-tuning of Llama needed? Would a base Llama-3.2 + RAG perform just as well?"
>
> The reviewer’s comment about ablation studies is valid. Since the initial manuscript was posted, we have added two additional user studies to help fill in the gaps: 1) a comparison of three possible “expert” stand-alone models: GPT-5, DeepSeek-R1-0528-Qwen3-8B, and LLaMA-TOMMI-1.0; 2) the platform with two possible “expert” models: OpenAI GPT-5, and LLaMA-TOMMI-1.0. These studies further emphasized that, while the platform RAG + synthesis agent does an excellent job of improving weaker answers, the fine-tuned expert model does a better job of aligning with the course material. We have added the additional results and discussion to the manuscript to emphasize.
>
> Updates made to paper; revised version to be uploaded shortly (Tables 3 and 5, with additional discussion in section 4.2).

---

### Review · Reviewer_AEqy · 2026-02-15

**Summary Of Contributions:**

This paper presents AI-U, a course-specific large language model system designed to support a graduate-level engineering course. The system integrates several components: automated Q&A generation from course materials, parameter-efficient fine-tuning (LoRA) of a domain-aligned expert model, retrieval-augmented generation (RAG) over course artifacts, and a reasoning-based synthesis model to produce grounded answers with citations to textbooks and lecture videos.

The work is positioned primarily as a systems contribution. It provides an end-to-end pipeline for building a course-aligned assistant and demonstrates its effectiveness through a combination of automatic metrics, human evaluations, and comparisons with a strong general-purpose model under comparable retrieval settings.

Strengths:
1. The system design is modular and well described, making the approach potentially reusable for other courses.
2. The evaluation is relatively thorough for an applied LLM systems paper, including multiple evaluation methods and instructor feedback.
3. The problem setting is clear and realistic, focusing on a concrete course and emphasizing grounded, citation-aware responses.
4. The paper presents a compelling case for open-source, locally deployable models in domain-specific educational settings.

Weaknesses:
1. The cost and scalability of data generation and human auditing are not fully characterized.
2. Because LLAMA-TOMMI-1.0 was trained specifically on the course materials, the comparison with GPT-5 may be biased toward "style alignment" rather than "intrinsic model quality". GPT-5’s perceived weakness (mathematical sophistication exceeding the course scope) might be a strength in different contexts.

**Audience:**

Yes

**Audience Explanation:**

The paper addresses an increasingly important topic: how to build domain-specific, grounded LLM systems that operate reliably in real-world environments. The design choices—parameter-efficient fine-tuning, structured retrieval, reasoning-based synthesis, and citation-aware interfaces—are broadly relevant beyond education and applicable to enterprise knowledge assistants, scientific copilots, and other grounded LLM systems.

In addition, the work provides a concrete and well-documented case study of deploying LLM technology in a realistic instructional setting, which is likely to be of interest to both researchers and practitioners in the TMLR community.

**Broader Impact Concerns:**

Overall, I do not see major ethical concerns beyond those already acknowledged, but a slightly deeper discussion of long-term educational risks would be beneficial.

**Claims And Evidence:**

Yes

**Claims Explanation:**

The authors provide convincing evidence through both quantitative metrics (86.02% winner rate for cosine similarity) and qualitative human reviews (55% instructor preference for the fine-tuned platform). The inclusion of a reasoning model to compensate for base model weaknesses is well-documented.

**Requested Changes:**

1. Cost-Benefit Analysis (Critical): Quantify the human effort involved in the manual audit and LaTeX conversion stages. This is necessary to validate the claim of "scalability" for new courses.
2. Detailed Error Analysis (Strengthen): Provide a categorical breakdown of the "failure cases" (the 7% of poor Q&A pairs). Discuss the risks of subtle errors in a learning environment and how the system might flag "uncertain" answers to the user.

---

> ### Author Response · Authors · 2026-02-26
> **Response to AEqy review**
>
> We thank the reviewer for their valuable comments and suggestions. We are uploading our updated manuscript, which includes improvements based on this review and the other feedback received so far.
>
> Regarding the requested changes:
>
> ------------
> > Cost-Benefit Analysis (Critical): Quantify the human effort involved in the manual audit and LaTeX conversion stages. This is necessary to validate the claim of "scalability" for new courses.
>
> Thank you for this suggestion. We have added details to the manuscript to highlight the manual, labor-intensive portion of the LaTeX conversion stages. The supporting code for the automated portions (e.g., API calls to an LLM for Q/A generation) will be released with the manuscript source code and can be utilized by instructors when applying the framework to their courses. We note that the time required for a “manual audit” of the Q/A pairs will highly depend on the specifics for each implementation.
>
> Please see additional text in Section 3.2 (first paragraph).
>
> ------------
> > Detailed Error Analysis (Strengthen): Provide a categorical breakdown of the "failure cases" (the 7% of poor Q&A pairs). Discuss the risks of subtle errors in a learning environment and how the system might flag "uncertain" answers to the user.
>
> This is an excellent suggestion. We have added the requested breakdown and additional discussion to the Technical Appendix.  In the Conclusion, we have raised the matter of subtle errors introduced to the learning process and how they may be countered.
>
> To illustrate how the system might flag “uncertain” answers, we have added instructions to the synthesis prompt in our demo (see Supplementary Material) to label each response with one of three confidence levels: High, Medium, or Low. Additional text provides the rationale for the provided confidence level.
>
> Please see additional text in Section 3.2 (last paragraph), Section 3.4, and Technical Appendix, Section C.